# Cold Exposure Rejuvenates the Metabolic Phenotype of *Panx1^−/−^* Mice

**DOI:** 10.3390/biom14091058

**Published:** 2024-08-25

**Authors:** Filippo Molica, Avigail Ehrlich, Graziano Pelli, Olga M. Rusiecka, Christophe Montessuit, Marc Chanson, Brenda R. Kwak

**Affiliations:** 1Department of Pathology and Immunology, Faculty of Medicine, University of Geneva, CH-1211 Geneva, Switzerland; avigail.ehrlich@unige.ch (A.E.); graziano.pelli@unige.ch (G.P.); olga@rusiecki.com.pl (O.M.R.); christophe.montessuit@unige.ch (C.M.); brenda.kwakchanson@unige.ch (B.R.K.); 2Geneva Center for Inflammation Research, CH-1211 Geneva, Switzerland; marc.chanson@unige.ch; 3Department of Cell Physiology and Metabolism, Faculty of Medicine, University of Geneva, CH-1211 Geneva, Switzerland

**Keywords:** adipocytes, browning, cold, thermogenesis, Pannexin1, mitochondrial genes

## Abstract

Pannexin1 (Panx1) ATP channels are important in adipocyte biology, potentially influencing energy storage and expenditure. We compared the metabolic phenotype of young (14 weeks old) and mature (20 weeks old) wild-type (WT) and *Panx1^−/−^* mice exposed or not to cold (6 °C) during 28 days, a condition promoting adipocyte browning. Young *Panx1^−/−^* mice weighed less and exhibited increased fat mass, improved glucose tolerance, and lower insulin sensitivity than WT mice. Their energy expenditure and respiratory exchange ratio (RER) were increased, and their fatty acid oxidation decreased. These metabolic effects were no longer observed in mature *Panx1^−/−^* mice. The exposure of mature mice to cold exacerbated their younger metabolic phenotype. The white adipose tissue (WAT) of cold-exposed *Panx1^−/−^* mice contained more small-sized adipocytes, but, in contrast to WT mice, white adipocytes did not increase their expression of Ucp1 nor of other markers of browning adipocytes. Interestingly, Glut4 expression was already enhanced in the WAT of young *Panx1^−/−^* mice kept at 22 °C as compared to WT mice. Thus, Panx1 deletion exerts overall beneficial metabolic effects in mice that are pre-adapted to chronic cold exposure. *Panx1^−/−^* mice show morphological characteristics of WAT browning, which are exacerbated upon cold exposure, an effect that appears to be associated with Ucp1-independent thermogenesis.

## 1. Introduction

The accumulation of adipose tissue results from an excess of energy intake over energy expenditure. White adipose tissue (WAT) stores caloric intake as triglycerides (TGs) and secretes adipokines like leptin and adiponectin, which regulate systemic metabolism and energy homeostasis [1,2]. WAT dysfunction is a hallmark of many diseases, including obesity, diabetes, and atherosclerosis. In contrast, brown adipose tissue (BAT) has a high catabolic and thermogenic activity primarily mediated by the mitochondrial uncoupling protein 1 (Ucp1) and allows for energy dissipation. A third type of fat cell was discovered to emerge in WAT after cold exposure, a process known as WAT browning [3]. These so-called beige adipocytes have characteristics similar to BAT such as the high expression of Ucp1 and thermogenesis, implying that beige adipocytes play an essential role in modulating energy metabolism, fat storage, and energy expenditure [4,5]. Apart from cold exposure, other factors such as sex, body composition, physical activity, and age influence adipocyte function and energy expenditure.

A role for membrane channels in the regulation of adipocyte biology is increasingly appreciated. The activation of the non-selective Ca^2+^ transient receptor potential vanilloid ion channels TRPV1 and TRPV3 has been shown to exert an anti-adipogenic role in vitro and in vivo [6,7], and the cold-induced activation of TRP melastatin receptor 8 promotes Ucp1 expression and thermogenesis [8]. Connexin43 gap junction channels were found to propagate the cold-induced sympathetic neuronal signal that mediates WAT browning within an adipocyte population [9]. Furthermore, connexin43 in adipocyte progenitors restrained energy expenditure and promoted fat accumulation [10]. Finally, Pannexin1 (Panx1) channel activity was shown to regulate insulin-stimulated glucose uptake in adipocytes, thereby contributing to the control of metabolic homeostasis [11]. The deregulation of insulin-Panx1-purinergic signaling in adipocytes has also been proposed to contribute to adipose tissue inflammation [12].

Panx1 belongs to a three-membered glycoprotein family that forms membrane channels at the cell surface and in intracellular compartments. Panx1 channels traffic to the plasma membrane, where they primarily function as ATP release channels but may also serve roles in the release or uptake of ions and small molecules up to 1 kDa [13]. Panx1-deficient, adipose-derived stromal cells exhibit accelerated adipogenic differentiation and fat accumulation [11,14]. Twelve-month-old Panx1-deficient mice were reported to have a greater fat mass and smaller lean mass than wild-type (WT) mice under a normal chow diet [14]. Moreover, atherosclerosis-prone, apolipoprotein E-deficient mice lacking Panx1 also displayed increased fat mass, particularly at the level of the visceral and subcutaneous WAT [15]. Together, these findings suggest a role for Panx1 channels in WAT accumulation and, consequently, in energy storage. In a recent study, Panx1 channel activation was found to increase thermogenic responses in brown adipocytes via β3-adrenergic receptor stimulation [16].

These observations raised the hypothesis that Panx1 channels are involved in the beneficial metabolic phenotype switch observed during prolonged cold exposure. To address this possibility, we systematically examined the whole-body metabolism in young and mature WT and *Panx1^−/−^* adult mice and determined the effect of Panx1 deletion on cold-induced alterations in metabolism, adipogenesis, and WAT morphology. We found that deleting Panx1 exacerbated the metabolic and phenotypic changes induced during cold exposure in WT mice, although these overall beneficial effects are distinct from those usually attributed to adipocyte browning.

## 2. Materials and Methods

### 2.1. Mice

*Panx1^−/−^* mice were backcrossed >10 generations with C57BL/6J mice (Charles River, France), and they were maintained as described previously [17]. WT (C57BL/6J) and *Panx1^−/−^* mice were housed under conventional conditions with a 12/12 h day/night cycle. All experiments were performed using 14 weeks old (young) to 20 weeks old (mature) adult male mice. Unless otherwise specified, the mice were fed a regular chow diet and tap water ad libitum. The experimental design and study timeline of temperature changes and the age of mice at tissue and blood sampling are provided in Figure 1. In brief, 14-week-old mice were weighed and allocated into three experimental groups, such that the average body weight per genotype was similar for the three groups. Mice were housed in pairs in a temperature-, light- and humidity-controlled (40%) thermostatic chamber (MEDI1300; Froilabo, Lyon, France) or Phenomaster (TSE Systems, Berlin, Germany). Indirect calorimetry was performed during one week at 22 °C on group 1 (G1), after which the mice were killed, and the tissue and blood samples were obtained. Mice from G2 remained at 22 °C and full monitoring for a total of 6 weeks was performed. Mice from G3 were exposed to a prolonged 4-week period of cold (6 °C). Prior to the cold exposure, mice underwent a 1-week adaptation to the thermostatic chamber or Phenomaster at 22 °C, followed by a week of acclimatization at 14 °C. Experiments in the Phenomaster metabolic screening platform allowed for respirometry (indirect calorimetry), food and drink monitoring, and an examination of locomotor activity during the full 6-week period. Each cage was considered as one pooled sample for these metabolic measurements. Energy expenditure was measured relative to body weight. Fatty acid (FA) oxidation was calculated using the following formula:FA oxidation = energy expenditure × [(1 − respiratory exchange ratio/0.3].

The number of mice used per group for in vivo measurements were adjusted to the absolute minimum obtained from prior power calculations. As specified under statistical analysis, we had no statistical outliers, and all measured values were taken into account.

Finally, it is increasingly recognized that Panx1 plays a different pathophysiological role in male and female mice [18,19,20]. Results obtained on male and female mice can thus not be pooled into the same experimental group. Although we recognize the importance of studying pathophysiological mechanisms in both males and females, this would have doubled the number of animals, and we have, therefore, chosen to use only same sex mice (males) for this in vivo study.

### 2.2. Body Temperature and Composition Measurements

The body temperature of all mice was measured daily with an infrared camera FLIR E60 at a distance of 40 cm perpendicular to the eye during the week of acclimatization at 14 °C and during the first week of housing at 6 °C. The body composition was assessed non-invasively with an EchoMRI 700 quantitative nuclear magnetic resonance analyzer. The data were presented as the percentage of total body weight.

### 2.3. Oral Glucose Tolerance Test (OGTT) and Insulin Tolerance Test (ITT)

Mice were fasted for 6 h. Blood was collected from a small incision at the tip of the tail before (t = 0) and 7.5, 15, 30, 60, 90, and 120 min after oral gavage with a bolus of D-glucose (1 g/kg body weight; Sigma, Buchs, Switzerland) or an intraperitoneal injection of insulin (0.75 U/kg body weight; Sigma). A Contour next ONE glucometer (Bayer, Leverkusen, Germany) was used to measure the blood glucose levels.

### 2.4. Tissue Collection

Mice were fasted for 6 h before being euthanized under anesthesia with an intraperitoneal injection of ketamine (100 mg/kg) and xylazine (10 mg/kg). Blood was collected by cardiac puncture, and serum was obtained after 15 min centrifugation at 2655 *g* at 4 °C. The length of the intestines was measured with a ruler by an investigator blinded for the genotype of the mice. The lengths of the small intestine (duodenum, jejunum, and ileum) and colon were measured separately. The epididymal WAT was harvested and weighed prior to being divided into two parts. One part was snap frozen for RNA extraction from the tissue. The second part was fixed in formol for 24 h and then embedded in paraffin for histological analysis.

### 2.5. Biochemical Analysis of Serum

A Cobas C111 analyzer (Roche Diagnostics, Rotkreuz, Switzerland) was used to measure total cholesterol, TG, and free fatty acid (FFA) levels in serum. Serum adipokine concentrations were determined using enzyme immunoassays for leptin (Quantikine ELISA mouse/rat leptin; R&D systems, Zug, Switzerland), ghrelin (rat/mouse ghrelin total ELISA kit; Millipore, Schaffhausen, Switzerland) or fibroblast growth factor 21 (FGF21) (Quantikine ELISA mouse/rat FGF21; R&D systems) according to the manufacturer’s instructions.

### 2.6. Histology and Image Analysis

Five μm thick paraffin sections of fat tissue were processed for histology. Standard hematoxylin and eosin staining was performed on rehydrated sections, and slides were mounted in Aquatex (Merck-Millipore, Schaffhausen, Switzerland). Images (20× magnification) were acquired using the Axio Scan.Z1 (Zeiss, Jena, Germany) automated slide scanner. The Adiposoft plugin of ImageJ was used to analyze the number and size of lipid droplets [21].

### 2.7. Mitochondrial Isolation

Mitochondria were extracted following a previously published protocol [22]. The epididymal WAT was collected and washed in a homogenization solution containing 250 mM sucrose, 10 mM HEPES, and 0.1 mM EGTA (pH 7.2). The tissue was minced on ice and homogenized in the same solution supplemented with 2% fatty acid (FA)-free bovine serum albumin using a Polytron homogenizer. Homogenates were centrifuged for 10 min at 8500× *g*, and lipids were removed by decantation. Pellets were resuspended in the homogenization solution and centrifuged for 10 min at 700× *g*. Supernatants were centrifuged at 8500× *g* for 10 min. Pellets containing mitochondria were washed by repeating the centrifugation step and were processed for RNA extraction.

### 2.8. RNA Extraction, Reverse Transcription, and Quantitative RT-PCR

The NucleoSpin kit (Oensingen, Switzerland) was used to extract total RNA from the epididymal WAT and mitochondria. The Quantitect Reverse Transcription kit (Qiagen, Hombrechtikon, Switzerland) was used for reverse transcription, and real-time PCR was performed with the ABI Prism StepOnePlus Sequence Detection System (Applied Biosystems, Waltham, MA, USA) using the Taqman Fast Universal master mix (Applied Biosystems). Probes and primers for mouse cell-death-inducing DNA fragmentation factor-alpha-like effector A (Cidea), carnitine palmitoyltransferase 1A (Cpt1A), Cpt1B, Panx1, Ucp1, and beta-2-microglobulin (B2m) were purchased from Applied Biosystems. The ΔΔCt method was used to calculate the relative mRNA concentrations, normalized to B2m expression.

### 2.9. Protein Extraction and Western Blotting

Proteins were extracted from the epididymal WAT of WT or *Panx1^−/−^* mice, and Western blots were performed as described previously [23]. Membranes were overnight incubated with anti-Glut4 antibody (1:1000, Cell Signaling, Boston, MA, USA) or anti-Ucp1 (1:1000, Cell Signaling) antibody at 4 °C. Beta-actin (1:1000, Sigma) was used as a loading control.

### 2.10. Statistical Analysis

Statistical analyses were performed with Graphpad Prism 9 software, and results were reported as mean ± SEM. The distribution of the data was assessed by the D’Agostino–Pearson Normality test or the Kolmogorov–Smirnov test. Two group comparisons were performed using two-tailed unpaired Student’s *t*-tests or Mann–Whitney U tests. Multiple groups comparisons were performed using the analysis of variance (ANOVA) with Bonferroni’s post-test or Kruskal–Wallis test with Dunn’s post-test, where appropriate. No statistical outliers were detected in any of the analyses performed. Statistical differences are indicated as * *p* ≤ 0.05, ^†^
*p* ≤ 0.01, and ^‡^
*p* ≤ 0.001.

## 3. Results and Discussion

### 3.1. Panx1 Deletion Enhances Glucose over Fat Metabolism in 14-Week-Old Mice

To detect metabolic alterations caused by Panx1 deletion, we first compared 14-week-old *Panx1^−/−^* male mice to WT male mice of the same age. These young *Panx1^−/−^* mice had a lower body weight than the WT mice (Figure 2A). While there was no difference in body weight between the WT and *Panx1^−/−^* mice at weaning, this difference became significant from 8 weeks onwards and remained up to 14 weeks of age (Appendix A). Similar to another line of Panx1-deficient mice in an earlier study [14], echoMRI revealed increased fat mass in the *Panx1^−/−^* mice (Figure 2B). However, the serum levels of total cholesterol, FFA, and TG did not differ between the mice of both genotypes (Figure 2C–E). Indirect calorimetry revealed increased energy expenditure in the *Panx1^−/−^* mice during their active phase (Figure 2F), which was consistent with a globally higher ambulatory, fine, or total movement during this period (Appendix A). Indirect calorimetry further revealed that the *Panx1^−/−^* mice consumed more food and water, which was most pronounced during the second half of their active period (Figure 2G,H). The differences in cumulative intake remained stable during the inactive period. We then tested if these differences in food consumption were related to the levels of secreted hormones. Ghrelin (Figure 2I) and leptin levels (Figure 2J) were not different between *Panx1^−/−^* and WT mice. Increased caloric uptake may, however, result in an increased absorptive capacity by the gut [24]. Indeed, the small intestines of the *Panx1^−/−^* mice were approximately 2 cm longer than the ones of the WT mice (Appendix A). The colon showed no difference in length between the two genotypes (Appendix A). When the *Panx1^−/−^* mice were compared to the WT mice during the active period, their VCO_2_ was increased while their VO_2_ was not (Appendix A), suggesting an increased anaerobic metabolism during which the produced lactic acid is neutralized by bicarbonate. No differences were observed between the two genotypes during the inactive phase. The respiratory exchange ratio (RER) in the Panx1-deficient mice remained near 1 during the active period, showing that carbohydrate was the primary source of energy during this phase, whereas the RER in the WT mice slightly fluctuated during the active period as expected (Figure 2K). FA oxidation was decreased in the *Panx1^−/−^* mice during their active period when compared to the WT mice, which is consistent with increased anaerobic metabolism (Figure 2L). The inversion of the RER and FA oxidation curves during the inactive period of the *Panx1^−/−^* and WT mice (Figure 2K,L) shows that the *Panx1^−/−^* mice favored the use of FA for their metabolic needs during their inactive period. Finally, the *Panx1^−/−^* mice had better tolerance to glucose at 7.5 to 30 min after an oral glucose challenge (Figure 2M), which may be explained, at least in part, by the higher serum levels of FGF21 in the *Panx1^−/−^* mice (Figure 2N). Indeed, the hepatokine FGF21 stimulates glucose uptake in adipocytes but not in other cell types, and this effect is additive to the activity of insulin [25]. Surprisingly, we observed a lower insulin sensitivity as measured by ITT (Figure 2O).

These findings suggest that, despite their increased fat mass, the young (14 week old) *Panx1^−/−^* mice exhibited enhanced energy expenditure, food intake, and glucose metabolism, paradoxically without gaining weight. As the *Panx1^−/−^* mice displayed a lower sensitivity to insulin, the higher tolerance of the *Panx1^−/−^* mice to glucose may result from stronger absorption by the longer-length intestine. It is worth mentioning, in this regard, that the overexpression and deletion of Panx1 have both been associated with insulin resistance. On the one hand, Panx1 expression is higher in the adipose tissue of obese humans and correlates with the degree of insulin resistance [11]. Panx1 deletion from adipocytes in mice, on the other hand, exacerbates diet-induced insulin resistance and is necessary for insulin-dependent glucose absorption in adipocytes [11]. Altogether, Panx1 deletion alters metabolism in young mice by favoring carbohydrate as an energy source during the active phase and FA oxidation during the resting period.

### 3.2. The Metabolic Phenotype of Panx1^−/−^ Mice Normalizes between 14 and 20 Weeks of Age

Mice are subject to metabolic changes during their developmental stages. As a reference point for comparison with metabolic changes induced by chronic exposure to cold (6 °C), we thus examined the whole-body metabolism in WT and *Panx1^−/−^* mice when they grew up from 14 to 20 weeks of age. The *Panx1^−/−^* mice gained nearly 2-fold more weight than the WT mice during these 6 weeks, thereby reducing the weight difference observed at 14 weeks between the two genotypes (Figure 3A,B). Likewise, the fat mass difference found at 14 weeks of age between both genotypes was no longer present at 20 weeks (Figure 3C). As expected, the lean mass was also comparable between the two genotypes at this age (Appendix A). The normalization of fat mass between genotypes was caused by a relative decrease in body fat in the *Panx1^−/−^* mice between 14 and 20 weeks of age, whereas the percentage of body fat in the WT mice remained stable (Figure 3D). Again, the serum levels of total cholesterol, FFA, and TG did not differ between the mice of both genotypes (Figure 3E–G). The *Panx1^−/−^* mice were more active during the second half of their active phase (Appendix A), which was associated with a tendency to higher energy expenditure (Figure 3H). Energy expenditure was similar in both genotypes during the inactive phase, when movement was minimal (Figure 3H and Appendix A). The enhanced food and water consumption of the *Panx1^−/−^* mice at 14 weeks was nearly cancelled at 20 weeks of age (Figure 3I,J), and no significant differences were observed in serum ghrelin and leptin levels between both genotypes (Figure 3K,L). However, the difference in the length of the small intestines between the *Panx1^−/−^* and WT mice observed at 14 weeks persisted at 20 weeks (Appendix A). As expected, no difference was observed in the length of the colon between both genotypes (Appendix A). The differences in the RER found at 14 weeks of age between the two genotypes were no longer observed at 20 weeks (Figure 3M). The normalization in the RER between both genotypes was mostly caused by an adaptation in the RER of the mature WT mice, which increased during the active phase and decreased during the inactive phase, whereas the RER of the *Panx1^−/−^* mice remained stable during the 14-to-20 weeks of age period. In contrast, FA oxidation remained more elevated in the resting *Panx1^−/−^* mice of 20 weeks old, whereas FA oxidation in the active period normalized to that observed in the WT mice (Figure 3N). Finally, the differences in glucose handling (GTT and ITT) observed between the *Panx1^−/−^* and WT mice at the age of 14 weeks were no longer present at 20 weeks of age (Figure 3O,P). The latter observations were associated with the lower circulating levels of FGF21 in the *Panx1^−/−^* mice compared with the WT mice (Figure 3Q). The changes in the RER of 14-to-20-week-old WT mice are common and have been previously described [26]. Both the 20-week-old WT and *Panx1^−/−^* mice used carbohydrate as the primary source of energy during the active period. However, the use of TG during the resting period remained more pronounced in the *Panx1^−/−^* mice, which may explain the general decline in fat mass observed at 20 weeks of age.

### 3.3. Cold Exposure Evokes Different Metabolic Changes in Panx1^−/−^ Mice and WT Mice

Cold exposure is known to modulate energy metabolism and fat storage [4,5]. To determine if Panx1 contributes to these changes, the *Panx1^−/−^* and WT mice were placed in a Phenomaster for 1 week at 22 °C followed by 1 week acclimatization at 14 °C before decreasing the temperature in the cages to 6 °C for 4 weeks. All mice survived and reached the age of 20 weeks at the end of this cold exposure period. The body temperature of each mouse during the period of acclimatization at 14 °C and during the first 72 h at 6 °C decreased similarly and was not different between both genotypes (Appendix A). In the WT mice, cold exposure prevented the weight gain normally observed between 14 and 20 weeks of age (Figure 4A,B) and induced an increase in fat mass during this period (Figure 4C,D). Interestingly, *Panx1^−/−^* mice were less sensitive to cold exposure. They kept gaining weight (Figure 4A,B) and showed almost no increase in fat mass (Figure 4C,D). The lean mass of the 20-week-old *Panx1^−/−^* mice was not different compared to the WT mice (Appendix A). Whereas total serum cholesterol levels remained similar between the *Panx1^−/−^* and WT mice (Figure 4E), the *Panx1^−/−^* mice had lower FFA and TG levels (Figure 4F,G), suggesting increased lipid metabolism. The *Panx1^−/−^* mice had a higher energy expenditure during cold exposure than the WT mice, both in the active and inactive periods (Figure 4H), although the peak of activity in the *Panx1^−/−^* mice was at the transition to the nocturnal period (Appendix A). The *Panx1^−/−^* mice consumed more food and water during the second half of their active period, whereas differences in cumulative intake remained stable during the inactive phase (Figure 4I,J). The increased food intake in the *Panx1^−/−^* mice was associated with an increase in serum ghrelin and a decrease in serum leptin levels (Figure 4K,L) and still by a longer small intestine (Appendix A). No difference in colon length was observed between the two groups of mice (Appendix A). In terms of energy source, the *Panx1^−/−^* mice maintained the use of carbohydrates for a longer time during the active phase, while they mobilized fat faster and more efficiently during the resting period (Figure 4M,N). VO_2_ and VCO_2_ remained higher during both periods in the *Panx1^−/−^* mice (Appendix A). Importantly, the 20-week-old *Panx1^−/−^* mice exposed to cold exhibited a similar glucose metabolism than the 14-week-old *Panx1^−/−^* mice, indicating a better tolerance to glucose (Figure 4O). During the cold exposure, the levels of serum FGF21 were very low and similar in the WT and *Panx1^−/−^* mice (Figure 4P). Cold exposure did not restore the difference in insulin sensitivity observed in the 14-week-old *Panx1^−/−^* mice (Figure 4Q). Thus, the cold-induced restoration of glucose tolerance in the *Panx1^−/−^* mice appears independent of FGF21 and may mostly rely on the ghrelin/leptin ratio.

These results indicate that in the absence of Panx1, mature adult (20-week-old) mice respond to cold exposure by adjusting their metabolism to that observed in younger (14-week-old) mice at 22 °C. In fact, the *Panx1^−/−^* mice behave like they are pre-adapted to cold exposure, suggesting that Panx1 acts as an integrator of metabolic regulation mostly by controlling the TG storage and use by white adipocytes. It is worth noting that the mice were housed in pairs for indirect calorimetry measurements. On the one hand, the paired housing of male mice may affect the activity level and food intake due to dominant and submissive behaviour. On the other hand, single housing has been shown to induce stress and anxiety in mice, which may also cause weight loss. For this reason, paired housing is recommended to reduce stress-induced variations in pharmacological studies using metabolic cages [27].

### 3.4. Cold-Induced WAT Morphological Changes Are Amplified in Panx1^−/−^ Mice

During their active period, the *Panx1^−/−^* mice showed a clear preference for carbohydrates over FA as an energy substrate when compared to the WT controls. The expression and function of the insulin-responsive glucose transporter Glut4 are required for glucose disposal into muscle and adipose tissue [28]. The downregulation of Glut4 has been associated with obesity and diabetes [29]. We determined Glut4 protein expression in the WAT of mice by Western blot. Interestingly, the 14-week-old *Panx1^−/−^* mice had higher Glut4 expression in the WAT than the WT mice of the same age (Figure 5A,C). Further development to 20 weeks of age had no effect on Glut4 expression in the adipose tissue from mice of both genotypes (Figure 5A,C). Whereas Glut4 expression was enhanced in the WT mice housed at 6 °C, cold exposure did not increase further the expression of this glucose transporter in the WAT of the *Panx1^−/−^* mice (Figure 5A,C). These observations suggest that *Panx1^−/−^* mice are predisposed for Glut4-dependent glucose uptake by white adipocytes and thus for fat storage. Higher adipocyte fat content may also explain the lower leptin release by the *Panx1^−/−^* mice exposed to cold. Previous studies showed increased glucose utilization and Glut4 expression in brown adipose tissue after cold exposure [30]. This is consistent with our observation that cold induced an increase of Glut4 expression in the WT mice. Interestingly, Glut4 expression was not changed in skeletal muscle from the WT and *Panx1^−/−^* mice regardless of their age and the temperature they were housed (Figure 5B,D), indicating that glucose transport was only affected in WAT. It is noteworthy that mice with adipose-specific Glut4 overexpression showed enhanced glucose tolerance, lower fasting glycemia, and increased body fat mass [31], which corresponds remarkably well to the metabolic phenotype observed in young (14-week-old) *Panx1^−/−^* mice. The importance of adipose tissue in regulating systemic insulin sensitivity and whole-body energy balance is increasingly appreciated [32] and prompted us to investigate potential morphological changes in the epididymal WAT of the WT and *Panx1^−/−^* mice. To this end, we compared the adipose tissue of 14-week-old and 20-week-old mice subjected or not to cold. Although no differences in the WAT weight was found between the different groups of mice (Figure 5E), microscopic examination revealed changes in the number and size of adipocytes (Figure 5F–H). Thus, the number of adipocytes in the WAT of *Panx1^−/−^* mice after cold exposure was higher than in the WT controls housed in the same conditions (Figure 5G), which suggests either an enhanced browning process or the induction of hyperplasia (generation of new adipocytes) in the *Panx1^−/−^* mice upon cold exposure. To determine if the increased number of adipocytes was accompanied by changes in their size, we categorized the adipocyte diameters in the WAT from each group of mice. The proportion of small (≤50 μm), medium (51–69 μm), large (70–89 μm), and very large (≥90 μm) adipocytes was determined. The cold-induced browning of WAT is known to reduce adipocyte diameter [33], which was also observed here for each genotype (Figure 5H). Interestingly, the *Panx1^−/−^* mice housed at 22 °C or at 6 °C clearly had a larger proportion of small adipocytes than WAT from WT mice under the same condition (Figure 5H). Our observations suggest that *Panx1^−/−^* mice exhibit WAT that is prone to browning, Panx1 deletion amplifying cold-induced morphologic changes in WAT.

### 3.5. Panx1^−/−^ Mice Do Not Increase Expression of Mitochondrial Oxidation Genes upon Cold Exposure

A hallmark of brown and beige adipocytes as compared to white adipocytes, in addition to their size differences, is the much higher density of mitochondria [34]. Mitochondria are the primary source of ATP generated via oxidative phosphorylation. We have recently found that Panx1 is expressed in subsarcolemmal mitochondria of ventricular cardiomyocytes, where it regulates mitochondrial respiration [35]. Interestingly, Panx1 mRNA expression was found in the mitochondria of the WAT of the WT mice, and its absence was confirmed in mitochondria isolated from the WAT of the *Panx1^−/−^* mice (Figure 6A). Panx1 expression in WAT mitochondria was decreased by 4 weeks of the cold exposure of the WT mice (Figure 6A). As expected, the cold exposure increased the expression of Ucp1 mRNA (Figure 6B) and protein (Figure 6C) as well as of Cidea (Figure 6D) mRNA in the WT mice, two gene markers of WAT browning [36]. Furthermore, Cpt1 expression, a mitochondrial enzyme responsible for the translocation of FA from the cytoplasm to the mitochondrial matrix, which has been associated with Ucp1 upregulation during mitochondrial uncoupling [36], was also increased (Figure 6E,F). Whereas the mRNA of the non-mitochondrial protein Cidea also increased upon cold exposure in the *Panx1^−/−^* mice, Ucp1 mRNA and protein were not increased upon cold exposure in these mice (Figure 6B–D). In addition, the expression of Cpt1a and Cpt1b mRNAs was not induced by cold exposure in the *Panx1^−/−^* mice (Figure 6E,F).

As the oxidation of FA is necessary for the upregulation of the mitochondrial thermogenesis-associated gene Ucp1 during the process of WAT browning [37], these results, together with the absence of FGF21 induction (Figure 4P), indicate that browning fat thermoregulation is altered in Panx1-deleted WAT. Panx1 critically regulates basal and maximal mitochondrial respiration in a high-energy-demanding cell type such as ventricular cardiomyocytes. Via a yet unknown mechanism, the altered mitochondrial function induced an accumulation of ATP in the cardiomyocyte mitochondria of the *Panx1^−/−^* mice, and the increased spare capacity of mitochondrial ATP resulted in decreased sensitivity to ischemia/reperfusion injury [35]. Here, we show the absence of the induction of the mitochondrial genes Ucp1 and Cpt1 in *Panx1^−/−^* white adipocytes in response to cold exposure (Figure 6) while they have an increased energy expenditure (Figure 4), suggesting a more efficient functioning of Panx1-deficient mitochondria under these conditions. Indeed, the *Panx1^−/−^* mice oxidized less FA in their active period when energy expenditure was maximal (Figure 4). Additionally, the increased energy expenditure and higher RER in the *Panx1^−/−^* mice may result from a combination of increased physical activity and compensatory mechanisms in non-adipose tissues.

A similar phenotype was observed in the young (14-week-old) *Panx1^−/−^* mice kept under regular conditions (Figure 2), which was associated with a better glucose tolerance. An improvement of glucose tolerance could be recovered in the older *Panx1^−/−^* mice upon cold exposure, further underlining the critical involvement of Panx1 in the regulation of metabolic processes. In the absence of Ucp1-dependent thermogenesis, Panx1-deficient mice may increase the combustion of fuels derived from diet or white fat lipolysis to maintain their body temperature, as recently shown in mice defective in brown fat lipolysis [38]. The differences in body composition and metabolic features between the WT and Panx1-deficient mice almost vanished with development. It is unlikely that these changes are simply due to genetic drift, as the *Panx1^−/−^* mice were backcrossed >10 generations with C57BL/6J mice. In addition, similar differences in fat mass and lean mass have also been reported in *Panx1^−/−^* mice of another origin (Genentech Inc., South San Francisco, CA, USA) than the ones used in this study and backcrossed onto a C57BL/6N background [14]. Nevertheless, whether these differences are already detectable during the neonatal period or present in females remains to be investigated.

The beneficial metabolic effects of Panx deficiency are pronounced in younger mice at regular housing conditions and appear in older mice only upon cold exposure. There is substantial evidence that aging impairs the formation and function of beige adipocytes and/or leads to a decline in the thermogenic capacity of adipocytes [39,40,41]. Mechanisms underlying such changes are starting to be delineated. They comprise diminished adipose precursor cell pool size and adipogenic potential, mitochondrial dysfunction, decreased sympathetic signaling, and altered paracrine and endocrine signals. Possibly, *Panx1^−/−^* mice may escape in part from this age-dependent adaptation. Further research is required to fully understand this mechanism.

Regular exposure to cold temperatures has been suggested as a potential anti-diabesity therapy [39,42]. Interestingly, a single nucleotide polymorphism (SNP) in the human Panx1 gene has been shown to predispose to the development of endothelial dysfunction with increasing body mass index [43]. As the Panx1-400A > C SNP affects the Panx1 channel functionality, our data call for clinical studies focused on the possible use of these variants in the prediction, and possibly delay, of early diabetic conditions such as metabolic syndrome.

## 4. Conclusions

Our data provide novel evidence for an involvement of Panx1 in WAT biology, which is schematically summarized in Figure 7. In response to chronic cold exposure, the WAT of the *Panx1^−/−^* mice showed an apparent browning morphology but failed to express typical markers like Ucp1 and Cpt1. Ucp1-independent thermogenesis has been identified and shown to involve a Ca^2+^-ATPase2b calcium cycling mechanism in beige adipocytes [44,45]. Future work using inducible and/or conditional knockouts in adipocytes could provide further insight into the role of Panx1 in Ucp1-independent thermogenesis.

## Figures and Tables

**Figure 1 biomolecules-14-01058-f001:**
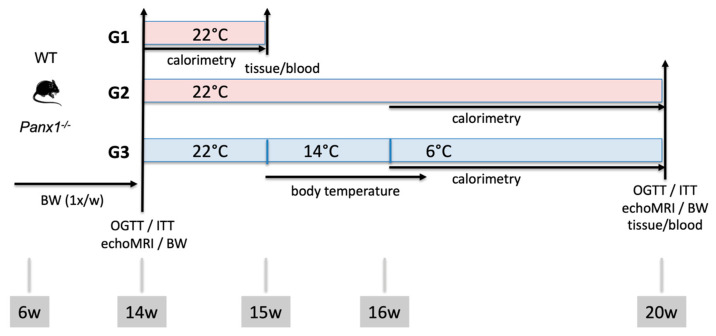
Experimental design and timeline of temperature changes. Fourteen-week-old mice were weighed and allocated into three experimental groups, such that, per genotype, the average body weight (BW) of the 3 groups was similar. Mice underwent echoMRI and were housed in pairs in a temperature-, light- and humidity-controlled thermostatic chamber or Phenomaster. Indirect calorimetry was performed during one week at 22 °C on group 1 (G1; top), after which the mice were killed, and the tissue/blood samples were obtained. Mice from G2 (middle) remained at 22 °C for a 6-week period, and calorimetry was performed for the last 4 weeks. Then, OGTT or ITT experiments were performed, echoMRI and body weight were measured, mice were killed, and tissue/blood samples were obtained. Mice from G3 (bottom) had one week of acclimatization at 14 °C and were subsequently exposed to a 4-week period of cold (6 °C). Then, OGTT or ITT experiments were performed, echoMRI and body weight were measured, mice were killed, and tissue/blood samples were obtained. Experiments in the Phenomaster metabolic screening platform allowed for respirometry (indirect calorimetry), monitoring of food and water consumption, and examination of locomotor activity. Each cage was considered as one pooled sample for these metabolic measurements.

**Figure 2 biomolecules-14-01058-f002:**
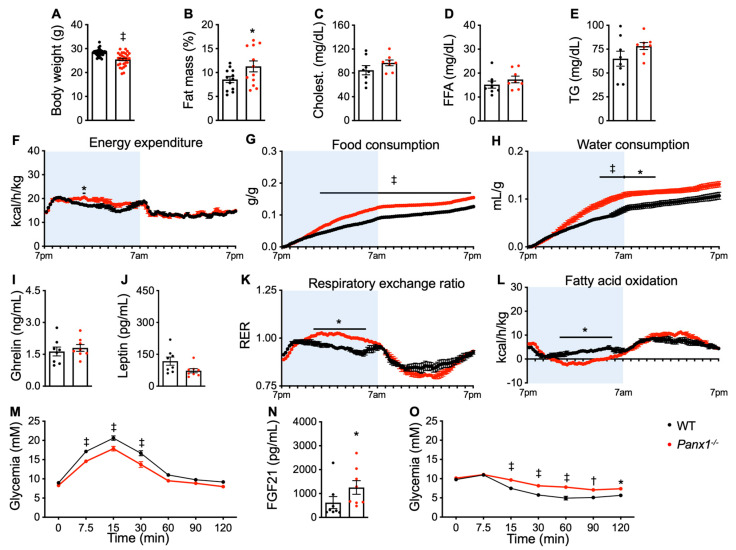
Panx1 deletion enhances glucose over fat metabolism in 14-week-old mice. (**A**) Body weights of all 14-week-old WT (black dots) and *Panx1^−/−^* (red dots) mice before separating them into 3 groups. (**B**) Fat mass of WT and *Panx1^−/−^* mice (G1) measured by echoMRI. Serum total cholesterol (**C**), FFA (**D**), and TG (**E**) levels in WT and *Panx1^−/−^* mice. Indirect calorimetry allowed the quantification of energy expenditure (**F**), cumulative food intake (**G**), cumulative water consumption (**H**), respiratory exchange ratio (**K**), and FA oxidation (**L**) in WT and *Panx1^−/−^* mice (mice were housed two per cage; each cage was considered as one pooled sample; N = 6 cages). The active period of mice is indicated in light blue. Ghrelin (**I**), leptin (**J**), and FGF21 (**N**) levels measured by ELISA in serum of WT and *Panx1^−/−^* mice. (**M**) Oral glucose tolerance test and (**O**) insulin tolerance test of WT (OGTT: N = 29; ITT: N = 20) and *Panx1^−/−^* (OGTT: N = 30; ITT: N = 20) mice. Data are shown as individual data points, except for time lines where the number of mice is specified, and expressed as mean ± SEM. * *p* ≤ 0.05, ^†^
*p* ≤ 0.01, and ^‡^
*p* ≤ 0.001.

**Figure 3 biomolecules-14-01058-f003:**
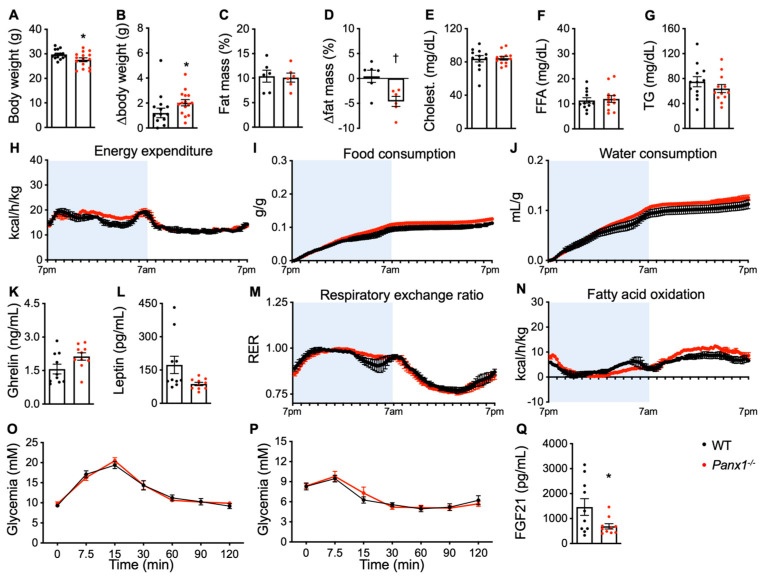
The metabolic phenotype of *Panx1^−/−^* mice normalizes between 14 and 20 weeks of age. (**A**) Body weight of 20-week-old WT (black dots) and *Panx1^−/−^* mice (red dots). (**B**) The weight gain of the mice between 14 and 20 weeks of age was also measured. (**C**) Fat mass and (**D**) difference in fat mass (as compared to their fat mass at 14 weeks) of WT and *Panx1^−/−^* mice were measured by echoMRI. Serum total cholesterol (**E**), FFA (**F**), and TG (**G**) levels in WT and *Panx1^−/−^* mice. Energy expenditure (**H**), cumulative food intake (**I**), cumulative water consumption (**J**), respiratory exchange ratio (**M**), and FA oxidation (**N**) in WT and *Panx1^−/−^* mice were measured by indirect calorimetry (N = 3 cages). The active period of mice is indicated in light blue. Ghrelin (**K**), leptin (**L**), and FGF21 (**Q**) levels measured by ELISA in serum of WT and *Panx1^−/−^* mice. (**O**) Oral glucose tolerance test and (**P**) insulin tolerance test of WT (OGTT: N = 15; ITT: N = 8) and *Panx1^−/−^* (OGTT: N = 15; ITT: N = 8) mice. Data are shown as individual data points, except for time lines where the number of mice is specified, and expressed as mean ± SEM. * *p* ≤ 0.05 and ^†^
*p* ≤ 0.01.

**Figure 4 biomolecules-14-01058-f004:**
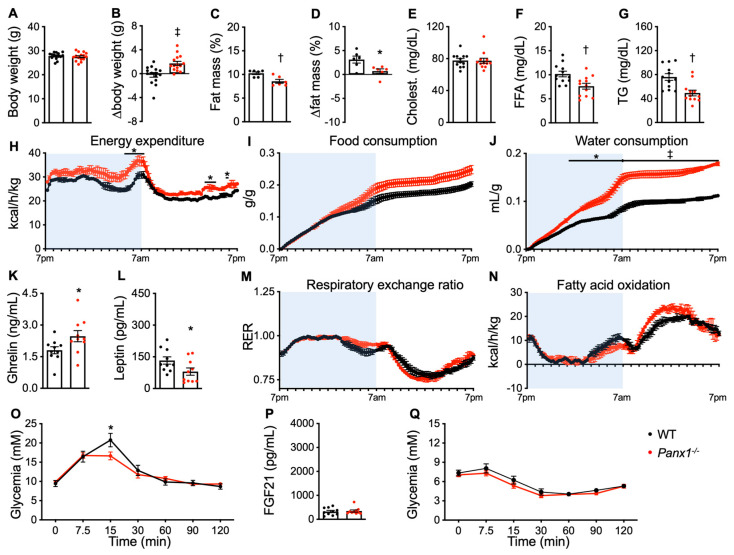
Cold exposure evokes different metabolic changes in *Panx1^−/−^* and WT mice. (**A**) Body weights of 20-week-old WT (black dots) and *Panx1^−/−^* (red dots) after 4 weeks of cold exposure at 6 °C. (**B**) The weight gain of the mice between 14 and 20 weeks of age was also measured. (**C**) Fat mass and (**D**) difference in fat mass (as compared to their fat mass at 14 weeks) of cold-exposed WT and *Panx1^−/−^* mice were measured by echoMRI. Serum total cholesterol (**E**), FFA (**F**), and TG (**G**) levels in cold-exposed WT and *Panx1^−/−^* mice. Energy expenditure (**H**), cumulative food intake (**I**), cumulative water consumption (**J**), respiratory exchange ratio (**M**), and FA oxidation (**N**) in WT and *Panx1^−/−^* mice were measured by indirect calorimetry (N = 3 cages) during cold exposure. The active period of mice is indicated in light blue. Ghrelin (**K**), leptin (**L**), and FGF21 (**P**) levels measured by ELISA in serum of cold-exposed WT and *Panx1^−/−^* mice. (**O**) Oral glucose tolerance test and (**Q**) insulin tolerance test of cold-exposed WT (OGTT: N = 15; ITT: N = 8) and *Panx1^−/−^* (OGTT: N = 15; ITT: N = 8) mice. Data are shown as individual data points, except for time lines where the number of mice is specified, and expressed as mean ± SEM. * *p* ≤ 0.05, ^†^
*p* ≤ 0.01, and ^‡^
*p* ≤ 0.001.

**Figure 5 biomolecules-14-01058-f005:**
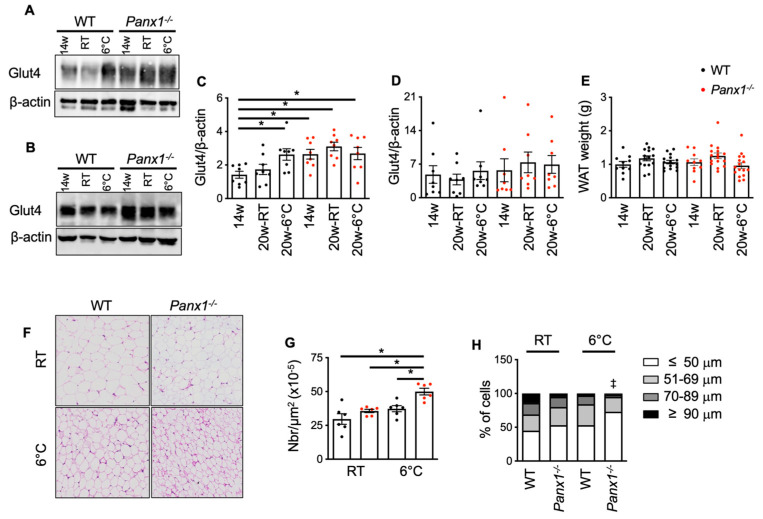
Cold-induced WAT morphological changes are amplified in *Panx1^−/−^* mice. (**A**,**B**) Illustrative Western blots (Western blot original images can be found in Appendix A) and (**C**,**D**) quantification of Glut4 expression in WAT (**C**) and skeletal muscle (**D**) from 14-week-old or 20-week-old WT and *Panx1^−/−^* mice housed either at 22 °C or at 6 °C during 4 weeks. Glut4 expression was normalized towards beta actin. (**E**) Weight of WAT pads from 14-week-old or 20-week-old WT and *Panx1^−/−^* mice at 22 °C or exposed to cold. (**F**) Hematoxylin and eosin staining on paraffin sections from WAT of WT and *Panx1^−/−^* mice at 22 °C or exposed to cold. Quantification of the numbers (**G**) of adipocytes in the WAT of WT and *Panx1^−/−^* mice at 22 °C or exposed to cold. Data are shown as individual data points and expressed as mean ± SEM. (**H**) Quantification of the proportion of small (≤50 μm), medium (51–69 μm), large (70–89 μm), and very large (≥90 μm) adipocytes in WAT of WT and *Panx1^−/−^* mice at 22 °C or exposed to cold. * *p* ≤ 0.05 and ^‡^
*p* ≤ 0.001.

**Figure 6 biomolecules-14-01058-f006:**
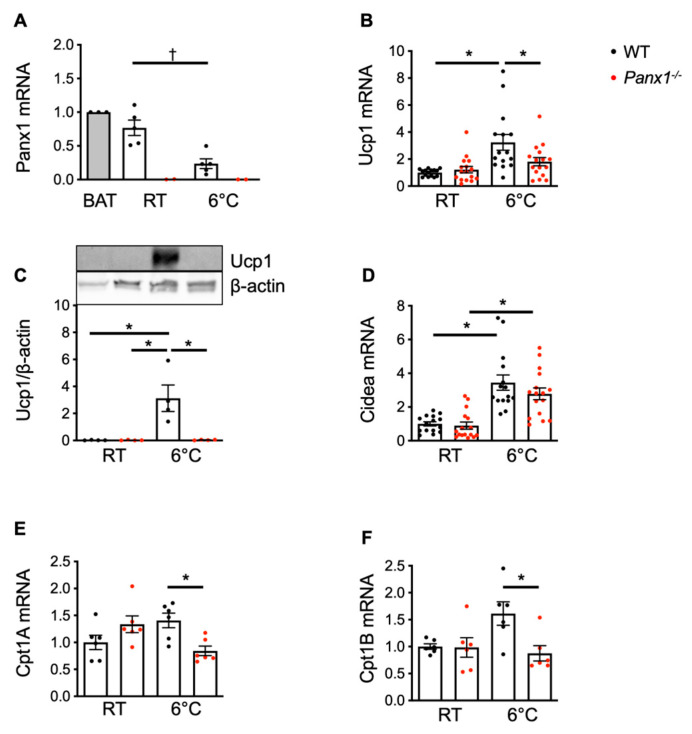
*Panx1^−/−^* mice do not increase expression of mitochondrial oxidation genes upon cold exposure. (**A**) Relative mRNA expression of Panx1 in mitochondria extracted from WAT of 20-week-old WT (white bar, black dots) and *Panx1^−/−^* (white bar, red dots) mice at 22 °C or exposed to cold (6 °C). BAT mRNA extracts from WT mice (grey bar) were used as positive control. Ucp1 (**B**) mRNA expression was established by qPCR in WAT from WT (black dots) and *Panx1^−/−^* (red dots) mice at 22 °C or exposed to cold. (**C**) Ucp1 protein expression was determined by Western blot (Western blot original images can be found in Appendix A). Ucp1 expression was normalized towards beta actin. Cidea (**D**), Cpt1a (**E**), and Cpt1b (**F**) mRNA expressions were assessed in WAT from WT (black dots) and *Panx1^−/−^* (red dots) mice at 22 °C or exposed to cold. mRNA expression of all investigated genes was normalized to B2m. Data are shown as individual data points and expressed as mean ± SEM. * *p* ≤ 0.05 and ^†^
*p* ≤ 0.01.

**Figure 7 biomolecules-14-01058-f007:**
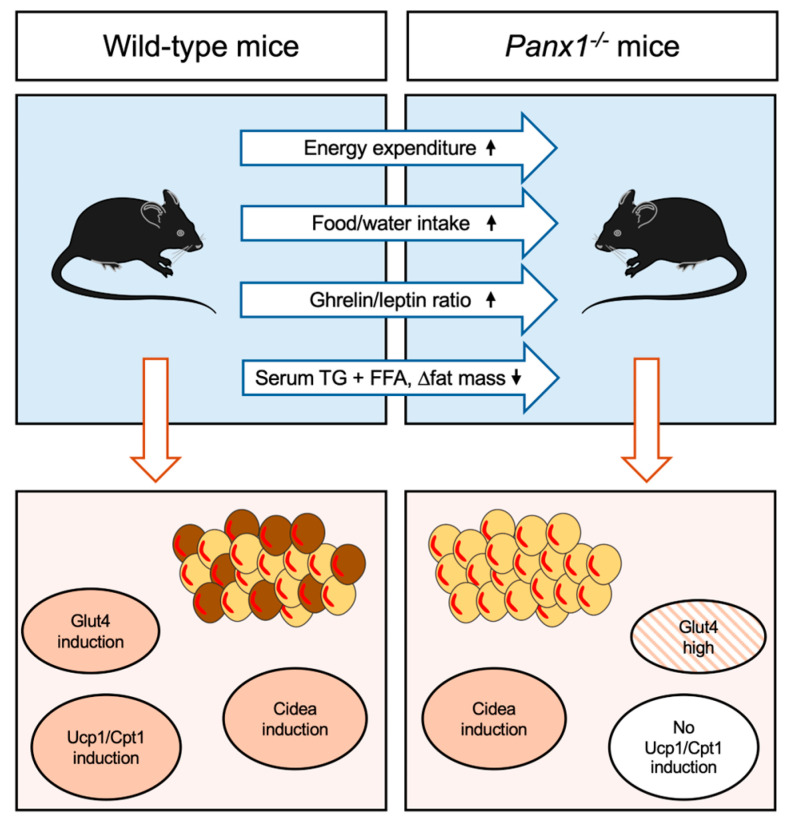
Schematic illustrating the main findings of our study. Upon chronic cold exposure (6 °C) *Panx1^−/−^* mice exhibit increased energy expenditure and have an increased ghrelin/leptin ratio resulting in increased food and water intake. Their serum TG and FFAs, as well as fat mass, are relatively decreased as compared to WT mice under the same conditions. WAT of cold-exposed *Panx1^−/−^* mice contain smaller adipocytes than WT mice with increased levels of Glut4. While chronic cold exposure induced Glut4, Cidea, Ucp1, and Cpt1 in WAT of WT mice, chronic cold exposure of *Panx1^−/−^* mice failed to induce Ucp1, Cpt1, and Glut4 in their WAT, pointing to Ucp1-independent thermogenesis in *Panx1^−/−^* mice.

## Data Availability

The data that support the findings of this study are available from the corresponding author upon reasonable request.

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
