# Peer review of "Cold Exposure Rejuvenates the Metabolic Phenotype of Panx1−/− Mice"

_biomolecules, 2024, doi:10.3390/biom14091058_

Round 1

Reviewer 1 Report

Comments and Suggestions for Authors

IEW

Introduction to the topic of the article

Maintaining a constant body temperature requires maintaining a balance between production

heat and its loss. Adipose tissue plays an important role in this. Adipose tissue is a connective tissue and comes in two varieties: yellow adipose tissue (Wat) and brown adipose tissue (Bat). the first one is made of single-vesicular fat cells (adipocytes) (containing one large drop of fat filling practically the entire cell), whose main task is to store fat and produce leptin; the second - from multi-vesicular adipocytes (containing many small fat drops), compared to yellow adipose tissue, brown tissue is very richly vascularized and innervated. its adipocytes are much smaller (14–40 μm compared to 10–150 μm in yellow fat), contain less fat (30 vs. 82%), the nucleus is located centrally and not peripherally, and, what is very important, they are characterized by significantly more mitochondria with large and numerous cristae. however, the most important feature is the presence of an uncoupling protein: UCP1 (uncoupling protein), or thermogenin, which is not found in any other tissue. Yellow and brown adipose tissue do not have to form separate clusters, but may be anatomically related. The number of Bat cells scattered between Wat cells depends not only on the species and age of the animal, but also on environmental conditions, diet and ambient temperature. Although pannexins were discovered 15 years ago as the "second family of gap junction proteins" in vertebrates, most connexin and innexin gap junction inhibitors were later found to also inhibit the Panx1 channel. Panx1 forms a patented, unconnected membrane channel that allows the exchange of molecules smaller than 1.5 kDa between the intra- and extracellular spaces. Six Panx1 subunits oligomerize to form a pannexon channel in the unattached plasma membrane. The properties of the channels are unusual and vary depending on the stimulation mode.

Title of the work

The title of the work reflects the scope of the research conducted.

Abstract

The abstract requires correction and supplementation with the most important results (numerical data) including statistical significance.

Introduction

The introduction to the research topic and the selection of literature are appropriate. The introduction should distinguish the research hypothesis and the purpose of the work

Material and methods

The studies were properly designed and the number of animals was appropriate. Please explain why only males were included in the study? Please explain the basis for the duration of the experiment? The procedure, type of biochemical and molecular measurements and analyzes were appropriate.

Results and discussion

The results section is presented in the form of 7 figures and descriptions. This chapter is very well presented and the descriptions of the results are sufficient. The discussion of the results is very well conducted, reflects the results of subsequent stages of the experiment and relates them to the results of other researchers.

Review Summary

In my opinion, the work presented for evaluation does not raise any major concerns, except for the explanations regarding the Materials and methods chapter. Once completed, the article is suitable for printing in the Biomolecules Journal.

Reviewer 2 Report

Comments and Suggestions for Authors

Please refer to the comments below.

1. For Figure 4 and related results, If there was no difference between lean mass and fat mass, what were the factors contributing to the weight gain?

2. For Figure 5 and related results, a mere occurrence of smaller adipocytes does not warrant browning phenotype. Have the authors tested the possibility of hyperplasia (generation of newer adipocytes usually associated with healthier phenotype)? Since UCP1 and Cidea are not upregulated, this claim of browning phenotype is not convincing. Other parameters such as mitochondrial respiration are not measured. 

3. Observations about Panx -/- mice in this study are in direct contradiction with the reference cited in the introduction section. The intro cites this reference which suggests Panx 1 activation leads to thermogenic phenotype. However, the authors have shown, although not convincingly, that the Panx deletion instead is beneficial for the mouse phenotype.

4. What were the reasons for higher energy expenditure and higher respiratory exchange rates if the browning genes and mitochondrial oxidation genes were not upregulated?

5. The authors need to include evidence of UCP1 independent thermogenesis in these mice. If it is absent, the authors need to provide satisfactory explanation about improved metabolic phenotype of Panx -/- mice.

6. The effect of Panx -/- seems to be more pronounced in younger mice in normal conditions and in older mice only in cold exposure, suggesting that the increasing age may have it's effects on the Panx -/- phenotype. Could the authors please comment in this?

Round 2

Reviewer 2 Report

Comments and Suggestions for Authors

The authors have satisfactorily addressed most of the previously mentioned concerns and comments.

This reviewer is still not convinced with the explanation about lack of Ucp1 expression and although induced, in fact lower Cidea expression upon cold exposure compared to WT. Furthermore, as the authors point out, they have demonstrated increased mitochondrial respiration in a different system i.e. cardiomyocytes. The adipocyte system may have a different pattern of regulation, and if there has been a published evidence in adipocytes about improved mitochondrial respiration, the authors may need to include it.
